# Mild Therapeutic Hypothermia Protects from Acute and Chronic Renal Ischemia-Reperfusion Injury in Mice by Mitigated Mitochondrial Dysfunction and Modulation of Local and Systemic Inflammation

**DOI:** 10.3390/ijms23169229

**Published:** 2022-08-17

**Authors:** Maxime Schleef, Fabrice Gonnot, Bruno Pillot, Christelle Leon, Stéphanie Chanon, Aurélie Vieille-Marchiset, Maud Rabeyrin, Gabriel Bidaux, Fitsum Guebre-Egziabher, Laurent Juillard, Delphine Baetz, Sandrine Lemoine

**Affiliations:** 1CarMeN Laboratory, Univ Lyon, INSERM, INRA, INSA Lyon, Université Claude Bernard Lyon 1, 69500 Bron, France; 2Hospices Civils de Lyon, Médecine Intensive Réanimation, Hôpital Edouard Herriot, 69003 Lyon, France; 3Hospices Civils de Lyon, Anatomopathologie, Groupement Hospitalier Est, 69500 Bron, France; 4Hospices Civils de Lyon, Néphrologie-HTA-Dialyse, Hôpital Edouard Herriot, 69003 Lyon, France; 5Hospices Civils de Lyon, Explorations Fonctionnelles Rénales, Hôpital Edouard Herriot, 69003 Lyon, France

**Keywords:** ischemia-reperfusion, mild therapeutic hypothermia, mitochondria, inflammation, acute kidney injury, chronic kidney disease, fibrosis

## Abstract

Renal ischemia-reperfusion (IR) injury can lead to acute kidney injury, increasing the risk of developing chronic kidney disease. We hypothesized that mild therapeutic hypothermia (mTH), 34 °C, applied during ischemia could protect the function and structure of kidneys against IR injuries in mice. In vivo bilateral renal IR led to an increase in plasma urea and acute tubular necrosis at 24 h prevented by mTH. One month after unilateral IR, kidney atrophy and fibrosis were reduced by mTH. Evaluation of mitochondrial function showed that mTH protected against IR-mediated mitochondrial dysfunction at 24 h, by preserving CRC and OX-PHOS. mTH completely abrogated the IR increase of plasmatic IL-6 and IL-10 at 24 h. Acute tissue inflammation was decreased by mTH (IL-6 and IL1-β) in as little as 2 h. Concomitantly, mTH increased TNF-α expression at 24 h. One month after IR, mTH increased TNF-α mRNA expression, and it decreased TGF-β mRNA expression. We showed that mTH alleviates renal dysfunction and damage through a preservation of mitochondrial function and a modulated systemic and local inflammatory response at the acute phase (2–24 h). The protective effect of mTH is maintained in the long term (1 month), as it diminished renal atrophy and fibrosis, and mitigated chronic renal inflammation.

## 1. Introduction

Renal ischemia-reperfusion (IR) injury can occur in various pathological situations such as resuscitated cardiac arrest, hemodynamic shock (e.g., sepsis, hemorrhage), surgery involving supra-renal aortic clamping (e.g., aortic aneurysm surgery), renal clamping (e.g., partial nephrectomy), or renal transplantation [1]. Renal IR can lead to acute kidney injury (AKI), increasing the risk of developing chronic kidney disease (CKD), and long-term mortality [2]. The prevention of renal IR injury is therefore crucial in all those pathological situations, to avoid short-term and long-term complications. The pathophysiology of renal IR shares common denominators including mitochondria-induced cell death, inflammation, and ultimately fibrosis [3]. It is therefore essential to better understand how to limit AKI and CKD in the context of renal IR.

Mild therapeutic hypothermia (mTH) is currently used in clinical practice for neuroprotection against IR injury [4], supported by clinical reports essentially in post-cardiac arrest care [5,6] and in infant hypoxic ischemic encephalopathy [7,8]. Regarding renal function, one randomized controlled trial showed that mild hypothermia (34 to 35 °C) in organ donors after declaration of brain death significantly reduced the rate of delayed kidney graft function among recipients, compared to normothermia (37 °C) [9]. A second clinical trial is currently ongoing to assess the nephroprotective effect of mild hypothermia in “expanded-criteria” brain-dead donors (ClinicalTrials.gov NCT03098706). However, the molecular and cellular mechanisms underlying this protection are not yet clearly defined.

It has been well-established that the mitochondrial permeability transition pore (mPTP) is a major determinant of IR injuries. The mPTP is therefore an interesting therapeutic target to protect from IR injuries [10]. Delaying its opening could be obtained by different ways and is related to reduced ischemia-reperfusion injury [11,12,13]. Indeed, previous works of our laboratory showed that hypothermia (34–35 °C) decreased the elevation of serum creatinine, i.e., AKI, and delayed the mPTP opening compared to the normothermic group in hypoxic cardiac arrest model in rabbits [9].

Renal IR is commonly associated with tissue and systemic inflammation [14]. Moreover, it has been shown that the inflammatory response to IR exacerbates the resultant injuries [15,16]. Different studies highlighted the importance of inflammation in the production of extracellular matrix and finally the onset of renal fibrosis and CKD, and especially the role of the balance between pro- and anti-inflammatory responses [17,18].

Here, the objective of our work was to test the efficiency of mild hypothermia conducted only during the short period of renal ischemia. We hypothesized that this stimulus would inhibit mPTP opening in parallel with a modulation of the inflammatory response that would protect kidney function and structure against IR in the acute phase of reperfusion and during the remodeling in the chronic phase.

Therefore, we conducted this study in mice with two different protocols: a bilateral renal IR model to assess acute injuries in the first 24 h, and a unilateral renal IR model, less severe to assess chronic lesions at 1 month. We aimed to confirm in those models that mTH (1) prevents AKI and CKD, (2) improves renal mitochondrial function, and (3) modulates the local and systemic inflammatory response in the short and long term.

## 2. Results

### 2.1. Mortality

Protocol 1, 20 min of bilateral renal ischemia, was a model of severe acute IR kidney injury, and it aimed at addressing the effect of mTH in the first 24 h of reperfusion. In Protocol 1, 3 mice out of 9 (33%) died in the IR-37 °C group, versus 1 mouse out of 9 (11%) in the IR-34 °C group during the 24 h of reperfusion (*p* = 0.58). No mortality was observed in the Sham-37 °C and Sham-34 °C groups during the 24 h of reperfusion. There was also no mortality in all the groups of mice that underwent 2 h of reperfusion.

Protocol 2, 15 min of unilateral renal ischemia followed by 1 month of reperfusion, was characterized by a limited mortality rate that allowed us to investigate the chronic effect of renal IR. None of the mice that underwent Protocol 2 died.

### 2.2. Renal Parameters

In the Sham animals in both Protocol 1 and 2, the temperature alone had no effect on any of the renal parameters: acute kidney injury, acute tubular necrosis, chronic kidney atrophy and fibrosis, and renal vascular resistance.

#### 2.2.1. Acute Kidney Injury (AKI), Tubular Necrosis (ATN) and Apoptosis

The results of Protocol 1 (20 min of bilateral renal ischemia followed by either 2 or 24 h of reperfusion) allowed us to assess acute effects and injury of IR on kidneys. Kidney function was evaluated by plasma urea assay. Plasma urea increased in the IR-37 °C group (18.7 mmol/L [17.3–19.0]) as soon as 2 h after reperfusion compared to Sham-37 °C (11.7 mmol/L [10.7–14.2]) (*p* = 0.02), and was even higher after 24 h of reperfusion (60.8 mmol/L [58.0–69.7] vs. 6.2 mmol/L [5.7–8.1] in IR-37 °C vs. Sham-37 °C respectively) (*p* = 0.004). When mTH was applied during ischemia, plasmatic urea concentration did not increase after 2 h of reperfusion (16.4 mmol/L [13.2–17.9]) when compared to Sham-34 °C (*p* = 0.06). Twenty-four hours after reperfusion, the plasmatic urea level concentration had significantly increased but remained lower in IR-34 °C mice than in IR-37 °C ones (*p* = 0.001) (Figure 1a,b).

Histological analysis of acute tubular necrosis was performed after 24 h of reperfusion. Renal IR led to significant ATN in both the IR-37 °C and IR-34 °C groups (score 2.23 [1.65–2.86] and 1.45 [1.16–1.85] respectively), as compared to their respective Sham-37 °C or -34 °C group (score 0.80 [0.65–0.85] and 1.00 [0.80–1.00] respectively) (*p* = 0.004 and *p* = 0.02 respectively). However, the histological score of ATN was significantly lower (−34%) in IR-34 °C when compared to IR-37 °C (*p* = 0.03) (Figure 1c,e).

Evaluation of acute tissue apoptosis was also performed by using a TUNEL assay after 24 h of reperfusion. Renal IR led to an increased number of apoptotic cells in the kidney in the IR-37 °C group (2.47% of all cells [1.88–3.31]) compared to Sham-37 °C (0.21% of all cells [0.10–0.38]) (*p* = 0.06), and apoptosis was significantly lowered when hypothermia was applied in IR-34 °C (0.41% of all cells [0.16–1.14]) compared to IR-37 °C (*p* = 0.03) (Figure 1d).

#### 2.2.2. Chronic Kidney Atrophy, Fibrosis and Apoptosis

In order to examine if mTH could affect the development of chronic kidney disease, we performed in Protocol 2 a 15-min unilateral renal ischemia followed by 1-month reperfusion. First, we evaluated the kidney atrophy by measuring the long axis of the ischemic kidney after 1 month of reperfusion. Ischemia induced a significant atrophy observed both in IR-37 °C and IR-34 °C groups (long axis 7.75 mm [7.5–8.0] and 8.5 mm [8.0 –9.0], respectively) when compared to their respective Sham-37 °C or Sham-34 °C (long axis 10.5 mm [10.0–10.75] and 10.25 mm [10.0–10.9], respectively) (*p* < 0.001 and *p* = 0.03 respectively). mTH in the IR-34 °C group was associated with a significant decrease of renal atrophy compared to IR-37 °C (*p* = 0.001). In our protocol, IR did not modify the contralateral control kidney lengths in IR-37 °C and IR-34 °C (where a possible compensative hypertrophy could have been expected) compared to the respective Sham, although the contralateral kidney in IR-37 °C were significantly bigger compared to those in IR-34 °C (Figure 2a).

CKD is commonly associated with renal fibrosis. Therefore, we quantified renal fibrosis in the ischemic kidney by Masson’s trichrome staining at 1 month of reperfusion. Accordingly, ischemic kidneys had developed fibrosis in the IR-37 °C group (27.7% of renal parenchyma [23.4–32.9]), significantly more extensive compared to the Sham-37 °C group (10.2% of renal parenchyma [7.3–13.8]) (*p* = 0.001). When mTH was applied, renal fibrosis was significantly less important in IR-34 °C (16.5% of renal parenchyma [8.8–18.8]) compared to IR-37 °C (*p* < 0.001), and it was not different from Sham-34 °C (9.5% of renal parenchyma [8.2–11.0]) (*p* = 0.18) (Figure 2b,e).

Eventually, we assessed apoptosis with a TUNEL assay in this chronic setting. IR was associated with a significant increase of tissue apoptosis in the IR-37 °C (7.22% of all cells [7.18–10.03]) and the IR-34 °C (4.78% of all cells [0.63–8.37]) groups compared to Sham (0.18% of all cells [0.05–0.28]) (*p* = 0.02 and *p* = 0.02 respectively) (Figure 2d).

#### 2.2.3. Renal Vascular Resistance (RVR)

Because our results showed a chronic renal dysfunction prevented by mTH, we then addressed if the renal vascular resistance (RVR) was modified early or chronically, in Protocol 2. Doppler US was used to follow non-invasively modifications in RVR. We showed a significant rise in RVR in ischemic kidneys at 20 min of reperfusion, in both IR-37 °C and IR-34 °C groups (Resistive Index 0.81 [0.75–0.86] and 0.83 [0.69–0.88] respectively), compared to their own baseline values measured a week before the procedure (Resistive Index 0.61 [0.58–0.67] *p* = 0.001, and 0.65 [0.60–0.69] *p* = 0.01 respectively). One month after reperfusion, these differences disappeared, and RVR in IR-37 °C and IR-34 °C were back to their baseline values (Resistive Index 0.63 [0.58–0.69] *p* = 0.67, and 0.68 [0.63–0.75] *p* = 0.10) (Figure 2c). There was no significant change of RVR in the Sham groups at neither 20 min nor 1 month of reperfusion, compared to their baseline.

### 2.3. Renal Mitochondria Function

Our hypothesis was that the beneficial effect of mTH on kidney function and structure could involve an early preservation of mitochondrial functions after 2 and 24 h of reperfusion. Temperature alone had no effect on renal mitochondrial functions between the 2 Sham groups in terms of calcium retention capacity (CRC) and oxidative phosphorylation (OX-PHOS) (Figure 3 and Figure 4).

#### 2.3.1. Mitochondrial Calcium Retention Capacity (CRC)

The CRC, measured on kidney mitochondria, was not altered by 20 min of bilateral ischemia followed by 2 h of reperfusion whatever the temperature used (respectively 280 nmol Ca^2+^/mg prot [220–380] at 37 °C and 280 nmol Ca^2+^/mg prot [220–420]) at 34 °C compared to their respective Sham conditions (320 nmol Ca^2+^/mg prot [280–640] and 280 nmol Ca^2+^/mg prot [260–440]) (*p* = 0.24 and *p* = 0.67)) (Figure 3a). Twenty-four hours after reperfusion, we observed a significant decrease of CRC in the IR-37 °C group (120 nmol Ca^2+^/mg prot [0–130]) compared to the Sham-37 °C group (320 nmol Ca^2+^/mg prot [280–360]) (*p* = 0.004). Conversely, the CRC was preserved by mTH in the IR-34 °C group 24 h after reperfusion (260 nmol Ca^2+^/mg prot [210–320]), compared to Sham-34 °C (280 nmol Ca^2+^/mg prot [240–340]) (*p* = 0.67) and reached a significantly higher value than in IR-37 °C group (*p* < 0.001) (Figure 3b).

#### 2.3.2. Oxidative Phosphorylation (OX-PHOS)

Electron transport chain (ETC) complexes I, II, and IV OX-PHOS activities in state 3, that is maximal ADP-stimulated mitochondrial respiration, were monitored in kidney mitochondria after 2 h or 24 h of reperfusion. After 2 h of reperfusion, OX-PHOS of complexes I, II, and IV were lowered in the IR-37 °C group, compared to Sham-37 °C (*p* < 0.05) (Figure 4a,c,e). At 24 h of reperfusion, only complexes II and IV OX-PHOS remained significantly altered in the normothermic IR condition compared to Sham-37 °C (*p* = 0.009 and *p* = 0.02 respectively) (Figure 4b,d,f). Conversely, mTH was associated with preserved complexes I, II, and IV OX-PHOS activities at 2 h and 24 h of reperfusion, as they were not different from Sham-34 °C (*p* > 0.05). Complex I activity in state 4 (oxygen consumption after ADP has been entirely phosphorylated) was altered 2 h after reperfusion in the IR-37 °C group when compared to Sham-37 °C (*p* = 0.008) (Appendix A). On the contrary, no modification was observed in the IR-34 °C group compared to Sham-34 °C (*p* = 0.55). The same isolated effect of IR at 37 °C was observed on Complex I respiratory coupling index (RCI), an index of mitochondria coupling, at 2 h of reperfusion (Appendix A). No effect of IR was seen at 24 h of reperfusion whatever the temperature (Appendix A).

### 2.4. Inflammatory Response

Temperature alone in the two Sham groups had no influence on the plasmatic concentrations of cytokines, nor on their mRNA expression in kidneys (Figure 5, Figure 6 and Figure 7). Because we have shown preservation of kidney function, structure, and mitochondrial function by mTH, we next studied whether this may have an impact on the systemic and local inflammatory response, in the acute and chronic setting of Protocol 1 and 2, respectively.

#### 2.4.1. Acute Systemic Plasmatic Inflammation

In Protocol 1, after 2 h of reperfusion, pro-inflammatory cytokine IL-6 level peaked after IR in both the IR-37 °C and IR-34 °C groups, without reaching significance but with a more pronounced trend for IR-37 °C versus Sham-37 °C (703 pg/mL [610–1107] versus 469 pg/mL [348–584], *p* = 0.06), rather than IR-34 °C versus Sham-34 °C (558 pg/mL [453–918] versus 408 pg/mL [301–477], *p* = 0.41) (Figure 5a). After 24 h of reperfusion, plasmatic IL-6 levels were lowered in all the groups but still significantly increased in IR-37 °C (100 pg/mL [55–161]) when compared to IR-34 °C (39 pg/mL [35–65]) (*p* = 0.03) (Figure 5b).

Plasmatic anti-inflammatory IL-10 level was not modified by renal IR after 2 h of reperfusion (Figure 5c). After 24 h of reperfusion, however, renal IR led to a significant elevation of plasmatic IL-10 only in the IR-37 °C group (713 pg/mL [347–1299]) when compared to Sham-37 °C group (*p* = 0.004), whereas mTH in IR-34 °C was associated with an undetectable plasmatic IL-10, as in the Sham 34 °C group (Figure 5d).

#### 2.4.2. Acute Renal Inflammation

We evaluated the local acute renal inflammation by the quantitative analysis of mRNA expression into the renal tissue at two different times of reperfusion. As for what was observed at the plasmatic level, bilateral renal IR induced a 12-fold overexpression of IL-6 mRNA after 2 h of reperfusion in the IR-37 °C group compared to Sham-37 °C (*p* = 0.008), which was lessened by half in the IR-34 °C group, making it non-significant when compared to Sham-34 °C (*p* = 0.06). Moreover, IL-6 mRNA increased expression was higher in the IR-37 °C group compared to the IR-34 °C group (approximately 2-fold, *p* = 0.008) (Figure 6a). After 24 h of reperfusion, IL-6 expression decreased in all the groups, and was still significantly elevated only in the IR-37 °C group (about 20-fold compared to Sham-37 °C, *p* = 0.004) (Figure 6b).

There was no significant modification of TNF-α expression after 2 h of reperfusion (Figure 6c). Nevertheless, at 24 h of reperfusion, mTH during renal IR induced a 3-fold increase of TNF-α mRNA expression in IR-34 °C when compared to IR-37 °C (*p* = 0.01) (Figure 6d).

Concerning IL-1β mRNA expression, renal IR induced a 1.6-fold elevation in the IR-37 °C condition when compared to Sham-37 °C group after 2 h of reperfusion (*p* = 0.03). mTH avoided modification of IL-1β expression (*p* = 0.10) (Figure 6e). After 24 h of reperfusion, no significant difference in IL-1β expression was observed between groups (Figure 6f).

Renal IR or mTH had no significant effect on mRNA expression of TGF-β, MCP1 or NFkB after 2 or 24 h of reperfusion (Appendix A).

#### 2.4.3. Chronic Renal Inflammation

In Protocol 2, after 1 month of reperfusion, unilateral renal IR was associated with a significant increase of IL-6, TNF-α, and IL-1β expression in the IR-37 °C and IR-34 °C groups compared to respective Sham (*p* < 0.05) (Figure 7a,c,e). Although TGF-β mRNA expression was increased only in IR-37 °C compared to Sham-37 °C, the use of mTH significantly attenuated the TGF-β mRNA expression compared to IR-37 °C group (*p* = 0.004) (Figure 7b). On the contrary, IR was associated with increased TNF-α mRNA expression by mTH when compared to normothermia (*p* = 0.009) (Figure 7c). MCP1 mRNA expression profile showed increased expression that reached significance only in the IR-34 °C group when compared to Sham-34 °C (*p* = 0.01) even if it seemed to have the same profile at 37 °C and did not reach significance, possibly because of an outlier (Figure 7f). Neither IR, nor mTH had any effect on NFkB expression in the chronic phase (Figure 7d).

We analyzed the macrophages infiltration by immunofluorescent F4/80 staining at 1 month of reperfusion. We showed that unilateral renal IR was associated with a significant infiltration of F4/80 positive cells in the ischemic kidney (IR-37 °C) compared to Sham-37 °C (respectively 2.58% [1.21–5.19] and 0.34% [0.08–0.75] of examined area) (*p* = 0.003). However, mTH prevented this macrophagic infiltration in IR-34 °C (0.74% [0.48–1.02] of examined area) compared to IR-37 °C (*p* = 0.005), and it was not different from Sham-34 °C (Figure 8).

## 3. Discussion

Our results demonstrate that mTH improves renal function and structure partly through a better mitochondrial function and a modulated local and systemic inflammation. Our present findings confirm that mTH allows renal protection on a daily or monthly scale.

We chose to apply mTH mostly during ischemia (per-conditioning) because it reflects more accurately clinical situations such as surgical procedures requiring a renal or aortic vascular clamping (i.e., partial nephrectomy, aortic aneurysms surgery, etc.) or kidney harvesting for transplantation, where mTH would be the easiest to implement in a daily clinical practice. Per-conditioning with mTH seems to be currently its most effective use against IR injuries [19]. In renal transplantation, one randomized controlled trial (RCT) has shown a benefit of mTH just before and during the kidney harvesting in brain-dead donors, by limiting the rate of delayed graft function in the first 7 days after transplantation, i.e., reperfusion [9]. Another RCT is currently ongoing with the same mTH applied on extended-criteria donors (ClinicalTrials.gov NCT03098706). In other settings, such as cardiac arrest or renal hypoperfusion (hemorrhage, sepsis, etc.) the onset of IR is not predictable, and mTH takes time to implement. In those situations, the effect of mTH as post-conditioning during reperfusion only is also currently challenged, as a recent RCT showed no benefit on mortality or neurological status of mTH maintained 24 h after resuscitation of cardiac arrest [20]. We performed a close monitoring of core body temperature (Appendix A) during the surgery and mTH was maintained about 10 minutes before the ischemia (the time it took for body temperature to stabilize before beginning the laparotomy) and up until the first minutes of reperfusion (as the mice were then woken up with a return to normothermia at the end of procedure), making it mostly per-conditioning but not strictly.

We chose a core body temperature of 34 °C based on previous data obtained in the laboratory [21]. mTH (32 to 36 °C) is indeed the easiest and fastest to implement, with less risk of life-threatening side effects. A body temperature lower than moderate therapeutic hypothermia (under 32 °C) is on the contrary associated with a potential risk of cardiac arrythmias and is used only in certain restricted situations (notably, aortic cross surgery). As for mTH, its main side effect is shivering in conscious patients, which is easily stopped when a sedation/anesthesia is used like in our model.

We made the choice to work with two different surgical models. Protocol 1, which consisted of 20-min bilateral renal ischemia followed by 2 or 24 h of reperfusion was severe enough to see features of AKI (urea increase and ATN combined with mitochondria dysfunction and increased inflammation), but too severe to keep mice alive after 24 h. That is why we had to perform a milder procedure, Protocol 2, to allow for the survival of mice, while CKD effectively developed (significative fibrosis combined with renal atrophy). Because these experimental protocols reproduced a good number of characteristics of the pathology studied, our models were pertinent, even if the use of two different models could nonetheless bring limitations. In Protocol 2, we only found a mild plasma urea elevation after renal IR, without differences between IR-37 °C and IR-34 °C (Appendix A). This was not surprising as the healthy contralateral kidney was expected to compensate through hyperfiltration the loss of function of the fibrotic kidney, making the plasma urea assay unreliable [22]. We cannot rule out differences in results and metabolic pathways involved between our 2 models caused by different protocols, like for example the compensation of the healthy contralateral kidney.

In Protocol 1, mTH applied during ischemia was beneficial on kidney function, assessed by plasmatic urea concentration and kidney structural damage, assessed by histology and acute tubular necrosis study, as well as apoptosis quantification by TUNEL. This is in line with previous work done by Xia et al. [23] in mice at 32 °C, where plasmatic creatinine and tubular apoptosis were significantly decreased by mTH. Nevertheless, they applied mTH much earlier as pre-conditioning, beginning 2 h before the procedure and during all the surgical time. Furthermore, the control group in this study was maintained at 38 °C, and one could object that this is not strictly equal to normothermia and could amplify deleterious effects of IR [24]. This study also reported reduced IR injury by mTH during the acute phase only; however very few data are available regarding long-term renal function after mTH in IR. A recent study demonstrated that mTH (body temperature at 33 °C, but the kidneys surface temperature was kept ex vivo at 24 °C) during renal ischemia was associated with less renal fibrosis, but estimated only 14 days after reperfusion [25]. We were able to show that the beneficial effect still exists 1 month after reperfusion. Renal function assessment is challenging in mice. Although we did not have enough blood to measure different biomarkers, we were able to show a significant increase of urea and most importantly, significant differences in histological lesions and apoptosis.

With renal artery Doppler US in our model, we reported that IR, independently of mTH, resulted in early increase of renal vascular resistance (RVR) 20 min after reperfusion. RVR are known to increase after a renal injury [26]. Moderate hypothermia (28 °C) itself has been shown to influence renal blood flow, glomerular capillary pressure and glomerular filtration rate, through an elevation of RVR [27]. This also shows that modifications of RVR can be monitored non-invasively through renal imaging, as soon as 20 min after reperfusion, and this could be repeated at other timepoints if necessary. At 1 month of reperfusion, however, there were no differences of RVR attributable to IR or mTH between the groups despite the differences of renal fibrosis. This suggests that RVR elevation only plays a role in the acute phase of the reperfusion, and not anymore as fibrosis has developed. Other parameters, such as renal perfusion estimated with contrast-enhanced ultrasound (CEUS) [28] or renal O_2_ saturation measured with photoacoustic imaging [29] have been proposed to monitor renal performance after IR and could be of interest.

Most of the hypothesis proposed to explain protection by mTH are based on the decrease of metabolic rate, the decrease of free radical production and the decrease of inflammatory processes. Therefore, we studied mitochondria, whose dysfunction is known to lead to production of free radicals and damage-associated molecular patterns (DAMPs), which are endogenous molecules that activates inflammatory response in case of cell damage or death. As expected, IR was associated with mitochondrial injury reflected by a significant decrease in CRC 24 h after reperfusion, and in OX-PHOS of ETC complexes activity 2 and 24 h after reperfusion in the normothermic group. mTH applied at the time of ischemia, was able to correct CRC defects, and to reestablish normal complex activity in our model. This is in line with results obtained during simulated ischemia in an in vitro model of isolated rat cardiomyocytes in which mild hypothermia (32 °C) was efficient in improving respiratory control ratio and reducing H_2_O_2_ production [30], and with results obtained in a model of rabbit cardiac arrest [21]. The decrease of mitochondrial ETC complexes activity was already observed in our previous works on kidney IR and heart IR, and actions which made it possible to improve renal or cardiac function after IR were always associated with preservation of mitochondrial function [12,13,21,31]. However, the exact mechanisms of mitochondrial protection involved in hypothermia remain unknown.

Surprisingly, renal IR induced a decrease in OX-PHOS of ETC complexes I, II, and IV at 2 h (protected by mTH), but at 24 h only an alteration of complexes II and IV persisted, and not of complex I. This might be partly explained by unexpectedly low values of complex I OX-PHOS in our Sham groups at 24 h. But it is also possible that mTH affects each ETC complex differently, as has been suggested with deep cold storage of rat kidneys at 4 °C before transplantation [32,33,34]. 

Moreover, it was shown that mitochondrial DNA (mtDNA) damage could be a hallmark feature of AKI, and mtDNA depletion has been found in preclinical model of AKI [35]. Therefore, it can be considered that mitochondrial dysfunction is associated with mitochondrial damage resulting in a loss of mtDNA, which could represent a stimulus in the development of an inflammatory response.

Because we thought that mTH could additionally act via a modulation of the post-ischemic inflammatory response, we assayed plasma IL-10 and IL-6 levels. Our results show, 24 h after reperfusion, that mTH limits the increase of plasma IL-6 and more surprisingly completely inhibits IL-10 plasmatic secretion induced by IR. In the renal tissue, IR also increased IL-6 mRNA, and mTH was associated with less IL-6 mRNA at 2 h, 24 h, and 1 month after reperfusion. mTH has already been associated with a decrease in IL-6 secretion in a model of cardiopulmonary resuscitation in pigs [36], in line with our results. In order to explain the mTH-mediated plasma IL-10 decrease, one could think that a significant limitation of the IR insult causes a diminished inflammatory response and therefore less IL-10 secretion. Indeed, in a model of canine acute respiratory distress syndrome, mTH was also associated with a preserved kidney function and a reduction of plasma IL-10 level [37]. However, in another study of experimental mice endotoxemia, mTH was related to a significant increase of both plasma IL-10 and IL-6 [38].

In line with the increase of IL-6 in the acute phase, we also observed, 2 h after reperfusion, a significant increase of IL-1β mRNA, certainly due to M1 macrophage activation [39], which was not present when mTH was applied. During the chronic phase, when fibrosis had already developed, IL-1β was elevated in both groups. We have not demonstrated any real effect of mTH on IL-1β chronic expression.

One study showed that the progression to CKD reflects failed or maladaptive epithelial repair leading to interstitial fibrosis [40]. TGF-β is a main regulator of renal fibrosis; therefore, it was important to consider its tissue level during reperfusion. TGF-β expression was not modified acutely as expected, but 1 month after IR, the amount of TGF-β mRNA was significantly higher in the normothermic group compared to those that had benefited of mTH. This result shows that about 20 min of mTH is able to change the TGF-β signaling pathway in the long term that usually conducts to fibrosis activation by overactivation of macrophages [39,41]. In accordance with these results, we found that renal IR led to a chronic macrophage (F4/80 positive cells) infiltration in the kidney 1 month after reperfusion, that was prevented when mTH was applied during ischemia.

MCP1 is a chemokine that regulates migration and infiltration of monocytes/macrophages. It is mainly associated with pro-inflammatory processes. In our model, IR in Protocol 1 did not modify acute MCP1 mRNA expression, but in Protocol 2, we observed a significant increase after 1 month only in the IR-34 °C group, although the lack of significance in the IR-37 °C group could be due to an outlier. This is quite unexpected, but in agreement with the results of a study in a murine ischemia-reperfusion hippocampal slice culture model. They showed an increased MCP1 expression in primary neurons treated with 10 μM cyclosporin A, which was even higher at 33.5 °C in the OGD/R injured group [42].

The different pathways described in the development of fibrosis are often associated with an increase of tissue TNF-α [43]. Surprisingly enough, in our model, mRNAs encoding TNF-α were more abundant in the mTH group than in the normothermic group as soon as 24 h of reperfusion in Protocol 1, and still more abundant even after 1 month in Protocol 2. TNF-α is commonly described as a pro-inflammatory cytokine and pro-apoptotic. Nevertheless, it was shown that neuroprotection conferred by cold preconditioning was mediated by an overexpression of TNF-α and IL-11 [44]. Consequently, it is therefore possible to think that in our model, TNF-α activation by mTH acts in the same way to protect the kidney, independently of the NFκB signaling pathway, whose mRNA expression was unchanged regardless of IR or temperature.

One could suggest that in our model, mTH exerted only an indirect effect on the inflammatory status and the cytokines expression and secretion. The protective effect of mTH (for example, through mitochondrial preservation) on cellular damage alone could have indeed lessened the triggering of inflammatory response by cell death. The modifications of expression of IL-6 and IL-1β could then just reflect the intensity of cellular damage due to renal IR, which was prevented by mTH. However, the unexpected finding of TNF-α overexpression by mTH supports the hypothesis mTH can also directly modulate the inflammatory response and immune cells, and modify cytokines expression. Further exploration will therefore be necessary to decipher the precise molecular or cellular pathways involved in this modulation of the inflammatory response and how it could participate to protection against IR.

In conclusion, we showed that mTH (34 °C) applied during a renal ischemia, alleviates renal dysfunction, histological damage, and tissue apoptosis, through a preservation of mitochondrial function and a modulated systemic and local inflammatory response at the acute phase (2 to 24 h after reperfusion) by decreasing systemic and local IL-6, by increasing TNF-α mRNA expression and by mitigating IL-1β mRNA expression. The protective effect of mTH is maintained in the long-term (1 month after reperfusion), as it diminishes renal atrophy and fibrosis, and still modulates the local renal inflammation, with mitigated TGF-β and activated TNF-α mRNA expressions, and with less macrophage infiltration in the kidney.

## 4. Materials and Methods

### 4.1. Animals and Surgical Procedures

Eight- to ten-week-old male C57BL6 mice (Charles River, France) were anesthetized by intraperitoneal injection of xylazine (5 mg/kg, Rompun; Bayer, Puteaux, France), ketamine (100 mg/kg, Imalgene 1000; Acyon, France), and buprenorphine (0.075 mg/kg, Vetergesic; Sogeval, Laval, France).

The animals were intubated and ventilated (Minivent, Harvard Apparatus; March, Germany), in order to perform the procedures described thereafter.

Core body temperature was maintained at either 34 °C or 37 °C during the whole surgery by using a rectal thermometer and a homeothermic pallet unit (PhysioSuite^®^ Kent Scientific, Torrington, CT, USA) (Appendix A).

All animal procedures were approved by local Ethics Committee (Claude Bernard Lyon 1 University, CEEA-55, Projet DR2019-09v2, APAFIS#19895-2019032121259377_v2) and in accordance with French and European Law.

### 4.2. Experimental Design

In Protocol 1 (2 h or 24 h of reperfusion), mice were assigned to 4 groups— normothermic Sham (Sham-37 °C) group (*n* = 5 with 2 h, *n* = 5 with 24 h), mTH Sham (Sham-34 °C) group (*n* = 5 with 2 h, *n* = 5 with 24 h), normothermic ischemia (IR-37 °C) group (*n* = 5 with 2 h, *n* = 9 with 24 h + 1 excluded mouse [see under]), and mTH ischemia (IR-34 °C) group (*n* = 5 with 2 h, *n* = 9 with 24 h + 1 excluded mouse [see under]) (Appendix A). All animals underwent laparotomy. Core body temperature was maintained at either 34 °C or 37 °C during the whole surgical procedure. The IR-34 °C group and IR-37 °C group underwent 20 min of bilateral renal ischemia by bilateral clamping of the renal vascular pedicles, by using 2 microvascular clamps simultaneously (Roboz Surgical Instruments, Washington, DC, USA). Ischemia and reperfusion were confirmed visually with the coloration of the kidneys. Sham procedures consisted of laparotomy only. All the mice were hydrated with a subcutaneous injection of 1.4% sodium bicarbonate (1 mL/mg of weight lost) immediately, 3 and 12 h after the surgical procedure. After 2 h or 24 h of reperfusion, blood samples were collected from the vena cava, and the kidneys were removed for mitochondria isolation, histology and qPCR.

In Protocol 2 (1 month of reperfusion), mice were assigned to 4 groups—normothermic Sham (Sham-37 °C) group (*n* = 5), mTH Sham (Sham-34 °C) group (*n* = 4), normothermic ischemia (IR-37 °C) group (*n* = 13), and mTH ischemia (IR-34 °C) group (*n* = 12) (Appendix A). The procedures were the same as in Protocol 1 except the ischemia. The IR-34 °C and IR-37 °C groups underwent 15 min of unilateral renal ischemia by selective clamping of the left renal vascular pedicle, with the right kidney left untouched. A renal ultrasound was performed with renal artery pulse-wave Doppler analysis on the left ischemic kidney, 1 week before the surgery (basal control), and 20 min and 1 month after the reperfusion. After 1 month of reperfusion, blood samples were collected, and the kidneys were removed for histology and qPCR. Both kidneys were measured ex vivo (maximal longitudinal length) by a blinded operator before sampling for conservation.

Two mice were excluded from Protocol 1: one in the IR-37 °C group because the renal ischemia was not obtained (as seen by the color of the kidneys remaining pink even after 20 min of bilateral clamping) and one in the IR-34 °C group because a homeothermic pallet unit failure maintained involuntarily the core body temperature above 37 °C resulting in an early post-operative death of the mouse secondary to hyperthermia and dehydration.

### 4.3. Renal Investigations

The operators performing all the procedures were blinded to the animals’ group assignment.

#### 4.3.1. Urea Titration

Acute kidney injury was assessed 2 and 24 h after reperfusion by measuring the plasmatic concentration of urea. Urea was measured by a kinetic UV urease method (Sobioda, Montbonnot-Saint-Martin, France) according to manufacturer’s instructions.

#### 4.3.2. Renal Histological Analysis

Renal tissue samples were fixed in formaldehyde 2 h, 24 h and 1 month after reperfusion, and embedded in paraffin.

To evaluate acute tubular necrosis, 4-μm sections were stained with periodic acid-Schiff reagents. Tubular injury was scored by a blinded pathologist semi-quantitatively who examined at least 10 fields, by using a scoring system ranging from 0 to 4. The lesions of ischemia were defined as a tubular dilatation, naked tubular basement membrane according to the following scoring system as described previously [12]: 0, no tubular injury; 1, less than 20% of tubules injured; 2, 21% to 50% tubules injured; 3, greater than 50% damage of tubule cells; 4, total destruction of all epithelial cells.

In order to assess chronic renal fibrosis 1 month after reperfusion, 4-μm sections were stained with Masson’s trichrome. Images were analyzed with Fidji software (ImageJ, NIH), with a self-made macro tool allowing the automatic quantification of fibrosis based on the ratio between green pixels (representing fibrosis) and all the pixels representing the whole kidney biopsy, expressed as a percentage.

Renal apoptosis was also quantified with terminal deoxynucleotidyl transferase dUTP nick-end labeling (TUNEL) staining at 24 h and 1 month of reperfusion. Four-μm sections were stained by using a commercial kit (Click-iT Plus TUNEL Alexa Fluor 594, Thermo Fisher Scientific Europe BV), according to manufacturer’s instructions. Images were obtained by using a 20× objective on a Zeiss Olympus BX63 microscope and analyzed with Fidji software (ImageJ, NIH). Results were expressed as a percentage of the total number of cells (stained with DAPI) per 20× magnification field.

At 1 month of reperfusion, renal tissue samples were also included in OCT and immediately frozen at −80 °C in order to perform latter immunofluorescence analysis.

### 4.4. Mitochondrial Investigations

The operators performing all the procedures were blinded to the animals’ group assignment.

#### 4.4.1. Preparation of Isolated Mitochondria

For mitochondrial preparation, a half-kidney was excised and immediately placed in cold isolation buffer (0.25 M sucrose, 0.1 mM ethylene glycol tetra-acetic acid, and 10 mM Tris, pH 7.4). The procedure was the same as was previously described [13]. We used 250-µg proteins for each CRC or respiration measurement.

#### 4.4.2. Calcium Retention Capacity

The calcium retention capacity (CRC), as originally described by Ichas et al. [45], represents the amount of Ca^2+^ necessary for the massive Ca^2+^ release by the mitochondria into the cytosol. Extra-mitochondrial Ca^2+^ concentration was measured with a Hitachi F2500 or F7000 spectrofluorometer as described before [13]. Ten nanomolar CaCl_2_ pulses were performed every minute, and the total amount of calcium required to open the mPTP was calculated and expressed in nmol Ca^2+^/mg mitochondrial protein. Representative recordings are shown in Appendix A.

#### 4.4.3. Mitochondrial Oxidative Phosphorylation (OX-PHOS)

Mitochondrial O_2_ consumption, expressed in nmol of oxygen/min/mg of mitochondria proteins, was recorded at 25 °C by using a Clark-type oxygen method electrode (Oroboros Oxygraph, Innsbruck, Austria), in order to determine sensitive rates of oxidative phosphorylation of electron transport chain (ETC) complex I, II, and IV as done before [13]. For each complex, state 3 represents the maximal oxygen consumption stimulated by adenosine diphosphate (ADP) when all required components are present. The lower oxygen consumption after complete ADP phosphorylation is considered as state 4. The ratio of state 3 to state 4 is named respiratory control index (RCI) and provides information about the mitochondrial coupling between respiration and phosphorylation. Representative recordings are shown in Appendix A.

### 4.5. Renal Ultrasound Imaging

Renal ultrasound imaging was performed in supine position by using a high-resolution ultrasonic imaging system (VEVO 3100 Fujifilm Visualsonics, Toronto, ON, Canada) under inhaled anesthesia with Sevoflurane, by an experienced operator blinded from the animals’ group assignment. A MX550D probe (Fujifilm Visualsonics, Toronto, ON, Canada) was used, fixed in place with an iron support. Initially, regular B-mode images were used to visualize the entire kidney and the renal artery, and to optimize imaging parameters for each animal.

A pulse-wave Doppler analysis was performed on the renal artery to obtain the systolic and diastolic velocity. The mean value of three distinct measurements was obtained for the peak systolic velocity and for the end diastolic velocity. The renal Resistive Index was then calculated as described before (RI = [peak systolic velocity − end diastolic velocity]/peak systolic velocity), as the reflect of renal vascular resistance and renal injury [26].

### 4.6. Inflammation Quantification

#### 4.6.1. Enzyme Linked Immunosorbent Assay (ELISA)

Circulating IL-6 and IL-10 concentrations were assessed on plasma by using enzyme-linked immunosorbent assay (Thermo Fisher Scientific, Waltham, MA, USA) according to manufacturer’s instructions.

#### 4.6.2. Quantitative Real-Time PCR (qRT-PCR)

The expression of IL-6, IL-1β, TNF-α, TGF-β, MCP1, and NFkB mRNA was assessed by quantitative reverse transcription-polymerase chain reaction (qRT-PCR). Prior to RNA isolation, snap frozen mouse kidney tissues were homogenized thanks to Precellys 24 (Bertin Instruments, Montigny-Le-Bretonneux, France). Total RNA was isolated from theses tissue homogenates, by using TRIzol (Ambion-Thermo Fischer Scientific, Illkirsh, France) according to the manufacturer’s protocol and reverse-transcribed by using a PrimeScript RT reagent kit (Takara Bio, Saint-Germain-en-Laye, France). Real-time PCR was performed by using the CFX96 real-time PCR system (Bio-Rad, Marnes-la-Coquette, France) and TB Green detection (Takara Bio, Saint-Germain-en-Laye, France) according to the manufacturer’s instructions. The relative quantitation of gene expression was calculated by using CFX Manager (Bio-Rad, Marnes-la-Coquette, France) thanks to the Pfaffl method [46]. Expression of the target genes was normalized to those of the β-actin gene (Actb). Primers were as follows: for IL-6 gene (Il6) forward 5′- AGT TGC CTT CTT GGG ACT GAT-3′ and reverse 5′- TCC ACG ATT TCC CAG AGA AC-3′; for IL-1β gene (Il1b) forward 5′- ACT GTT CCT GAA CTC AAC TG-3′ and reverse 5′- CTT GTT GAT GTG CTG CTG CG-3′; for TNF-α gene (Tnf) forward 5′- CCT CAC ACT CAG ATC ATC TTC-3′ and reverse 5′- TGG CAC CAC TAG TTG GTT GTC-3′; for TGF-β (Tgfb1) forward 5′- CCT GAG TGG CTG TCT TTT GA -3′ and reverse 5′- CGT GGA GTT TGT TAT CTT TGC TG -3′; for MCP1 (Ccl2) forward 5′-GTC CCT GTC ATG CTT CTG G-3′ and reverse 5′-GCT CTC CAG CCT ACT CAT TG-3′; for NFkB gene (Nfkb1) forward 5′-AGG CTT CTG GGC CTT ATG TG-3′ and reverse 5′-TGC TTC TCT CGC CAG GAA TAC-3′; and for Actb forward 5′-ACC TTC TAC AAT GAG CTG CG-3′ and reverse 5′-CTG GAT GGC TAC GTA CAT GG-3′.

#### 4.6.3. Immunofluorescence

Immunofluorescence was performed on 8 µm-sections of frozen renal tissue samples. Immunostaining was performed by using rat anti-mouse F4/80 antibodies (CI-A3-1: NB600-404, Novus Biologicals–Bio Techne SAS, Noyal-Chatilon-sur-Seiche, France) at 1:100 dilution. The secondary antibody used at 1:300 was Alexa Fluor 594-conjugated chicken anti-rat IgG (A21471, Thermo Fisher Scientific, Illkirsh, France). Briefly, the sections were fixed for 20 min with 3.7% formaldehyde, post-fixed with 0.1% Triton PBS solution, blocked with 2% goat serum for 30 min, and then incubated overnight with the primary antibody at room temperature. The sections were washed 3 times for 10 min each with PBS, followed by incubation for 30 min with the secondary antibody at room temperature. The sections were washed again with PBS, and nuclei were stained with DAPI. Images were obtained by using a 20× objective on a Zeiss Olympus BX63 microscope and analyzed with Fidji software (ImageJ, NIH, Bethesda, MD, USA). Results were expressed as a percentage of kidney tissue area per 20× magnification field.

### 4.7. Statistical Analysis

Data are expressed as median [with interquartile range]. All groups were compared with a Kruskal–Wallis test. Comparisons between 2 groups was performed using Mann–Whitney tests. Paired data were compared with Wilcoxon matched-pairs signed rank test when appropriate (Graphpad software 8.4.2, San Diego, CA, USA). *p* value less than 0.05 was considered significant.

## 5. Conclusions

We showed that mTH (34 °C) applied during a renal ischemia, alleviates renal dysfunction and histological damage, through a preservation of mitochondrial function and a modulated systemic and local inflammatory response at the acute phase (2 to 24 h after reperfusion) by decreasing systemic and local IL-6, by increasing TNF-α mRNA expression and by mitigating IL-β1 mRNA expression. The protective effect of mTH is maintained in the long-term (1 month after reperfusion), as it diminishes renal atrophy and fibrosis, and still modulates the local renal inflammation, with mitigated TGF-β and activated TNF-α mRNA expressions.

## Figures and Tables

**Figure 1 ijms-23-09229-f001:**
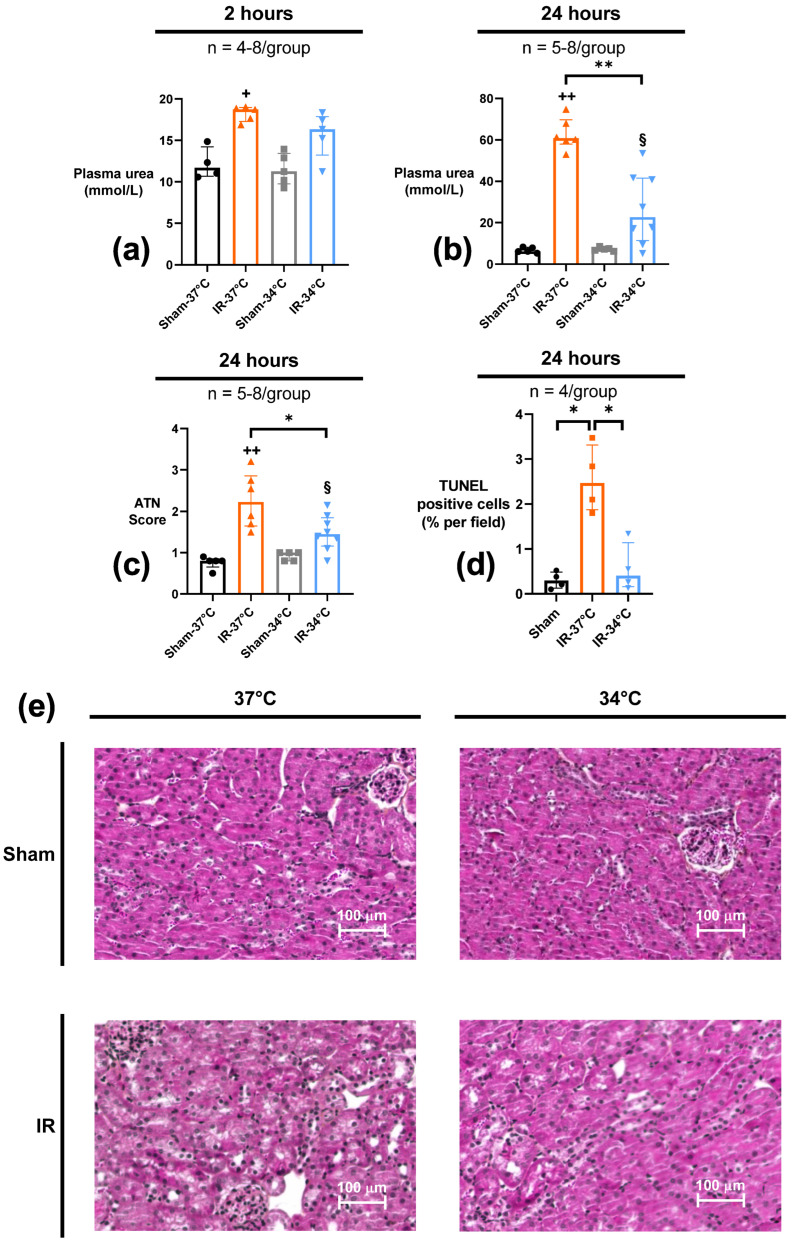
Renal functional parameters after 20 min of bilateral renal ischemia where the mice body temperature was maintained at normothermia (IR-37 °C) or mild therapeutic hypothermia (mTH) (IR-34 °C) and 2 h or 24 h of reperfusion, or after a Sham procedure with the same body temperature (Sham-37 °C and Sham-34 °C, or pooled as Sham). Renal function was assessed with plasma urea measured by a kinetic UV urease method after 2 h (**a**) or 24 h (**b**) of reperfusion. Renal ischemia-reperfusion induced acute kidney injury in IR-37 °C group at 2 h and 24 h, but mTH during ischemia (IR-34 °C group) attenuated this increase in plasma urea. (**c**) Renal acute tubular necrosis (ATN) was assessed at 24 h of reperfusion by a blinded pathologist. Histology scoring ranking from 0 to 4 was performed with: 0, no tubular injury; 1, less than 20% of tubules injured; 2, 21% to 50% tubules injured; 3, greater than 50% damage of tubule cells; 4, total destruction of all epithelial cells. mTH significantly lowered ATN score after ischemia-reperfusion, compared to normothermia. (**d**) Renal tissue apoptosis was assessed with TUNEL staining after 24 h of reperfusion. Renal ischemia-reperfusion induced a significant increase of cell apoptosis in the acute phase of reperfusion, that was reduced by mTH. (**e**) Representative images of 4-μm sections of kidney biopsy for each group, stained with periodic acid-Schiff reagents to assess acute tubular necrosis. Data are shown as median with interquartiles. * *p* < 0.05, ** *p* < 0.01, Mann–Whitney test. + *p* < 0.05, ++ *p* < 0.01 vs. Sham-37 °C, Mann–Whitney test. § *p* < 0.05 vs. Sham-34 °C, Mann–Whitney test.

**Figure 2 ijms-23-09229-f002:**
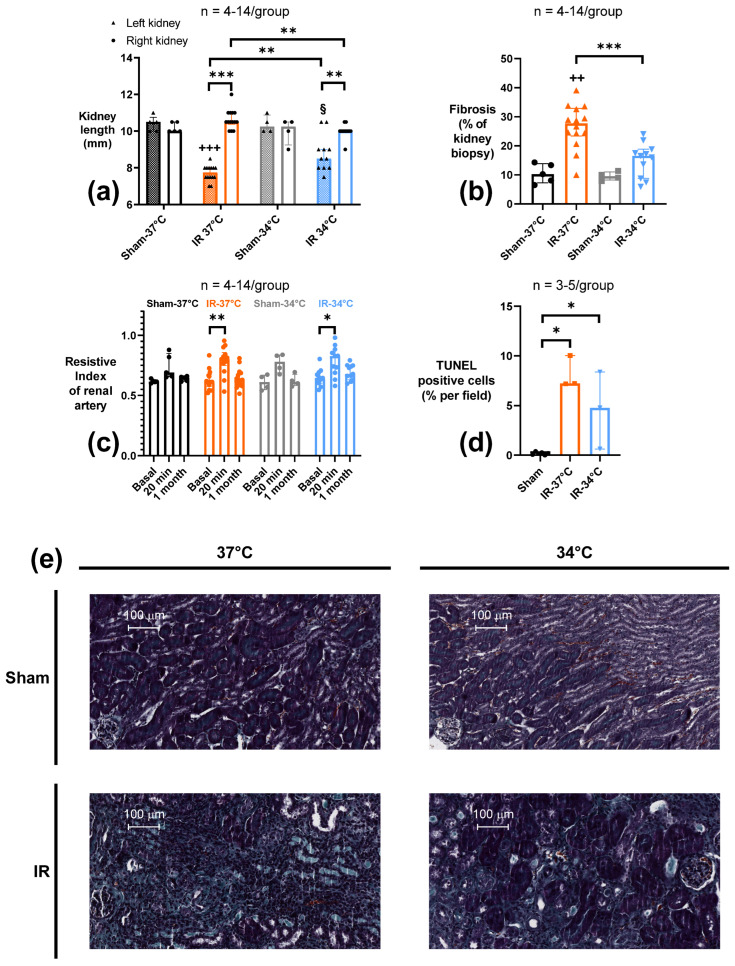
Renal functional parameters measured after 15 min of unilateral renal ischemia where the mice body temperature was maintained at normothermia (IR-37 °C) or at mild therapeutic hypothermia (mTH) (IR-34 °C) followed by 1 month of reperfusion, or after a Sham procedure with the same body temperature (Sham-37 °C and Sham-34 °C). (**a**) Kidney length (long axis) was measured ex vivo to assess renal size modification in the ischemic kidney and the contralateral one. Ischemia-reperfusion induced renal atrophy of the ischemic kidney in both IR-groups, but ischemic kidneys remained bigger. (**b**) Histology with Masson’s trichrome staining was used to quantify chronic renal fibrosis, through a computerized automatic quantification of fibrosis based on the ratio between green pixels (representing fibrosis) and all the pixels representing the kidney biopsy, and expressed as a percentage. Ischemia-reperfusion led to the development or extensive renal fibrosis, but mTH significantly alleviated it. (**c**) Renal vascular resistance was evaluated by the Resistive Index measured in the renal artery with pulse-wave Doppler ultrasound. Three measures were performed on the ischemic kidney, at baseline (7 days before surgery), and 20 min, and 1 month after reperfusion. Ischemia-reperfusion was associated with a rise in renal vascular resistance as soon as 20 min after reperfusion in both IR-37 °C and IR-34 °C groups compared to their own baseline values, but there was no observable difference after 1 month of reperfusion. (**d**) Renal tissue apoptosis assessed with TUNEL-staining after 15 minutes of unilateral renal ischemia where the mice body temperature was maintained at normothermia (IR-37 °C) or at mild therapeutic hypothermia (mTH) (IR-34 °C), followed by 1 month of reperfusion, or after sham procedure (Sham-37 °C and Sham-34 °C pooled as Sham). Renal ischemia-reperfusion induced a significant increase of cell apoptosis whatever the temperature used during ischemia. (**e**) Representative images of 4-μm sections of kidney biopsy for each group, stained with Masson’s trichrome, with collagen fibers, i.e., mostly fibrosis, stained in green. Data are shown as median with interquartiles. * *p* < 0.05, ** *p* < 0.01, *** *p* < 0.001, Mann–Whitney test or Wilcoxon test if appropriate. ++ *p* < 0.01, +++ *p* < 0.001 vs. Sham-37 °C, Mann–Whitney test. § *p* < 0.05 vs. Sham-34 °C, Mann–Whitney test.

**Figure 3 ijms-23-09229-f003:**
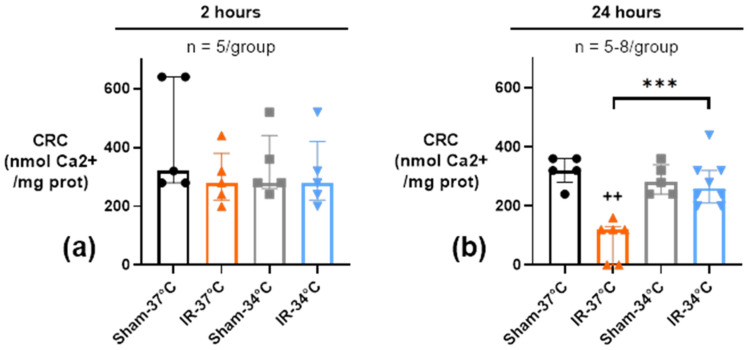
Calcium retention capacity (CRC) in kidney mitochondria after 20 min of bilateral renal ischemia where the mice body temperature was maintained at normothermia (IR-37 °C) or at mild therapeutic hypothermia (mTH) (IR-34 °C), and 2 h (**a**) or 24 h (**b**) of reperfusion, or after a Sham procedure with the same body temperature (Sham-37 °C and Sham-34 °C). Renal ischemia-reperfusion did not have any effect on CRC 2 h after reperfusion, but CRC was significantly altered at 24 h, whereas mTH prevented this decrease and preserved CRC. Data are presented as median with interquartiles. *** *p* < 0.001, Mann–Whitney test. ++ *p* < 0.01 vs. Sham-37 °C, Mann–Whitney test.

**Figure 4 ijms-23-09229-f004:**
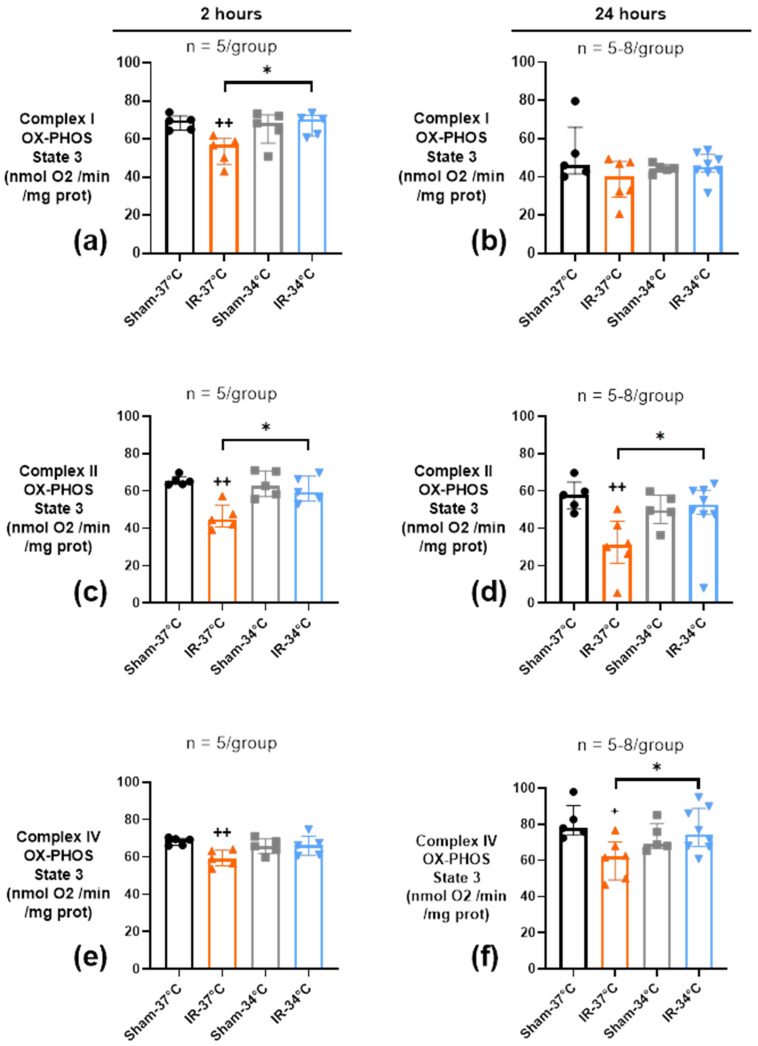
Oxidative phosphorylation (OX-PHOS) of kidney mitochondria after 20 min of bilateral renal ischemia where the mice body temperature was maintained at normothermia (IR-37 °C) or at mild therapeutic hypothermia (mTH) (IR-34 °C), and 2 h (**a**,**c**,**e**) or 24 h (**b**,**d**,**f**) of reperfusion, or after a Sham procedure with the same body temperature (Sham-37 °C and Sham-34 °C). Sensitive rates of OX-PHOS of electron transport chain (ETC) complex I, II, and IV were measured through O_2_ consumption of 250 µg kidney mitochondria. For each complex, state 3 represents the maximal oxygen consumption stimulated by adenosine diphosphate (ADP) when the substrates of the different complexes are present. OX-PHOS of complexes I, II, and IV were altered as soon as 2 h after reperfusion, and only OX-PHOS of complex II and IV remained altered after 24 h of reperfusion, compared to Sham. mTH prevented those alterations at 2 h and 24 h of reperfusion. Data are shown as median with interquartiles. * *p* < 0.05, Mann–Whitney test. + *p* < 0.05, ++ *p* < 0.01 vs. Sham-37 °C, Mann–Whitney test.

**Figure 5 ijms-23-09229-f005:**
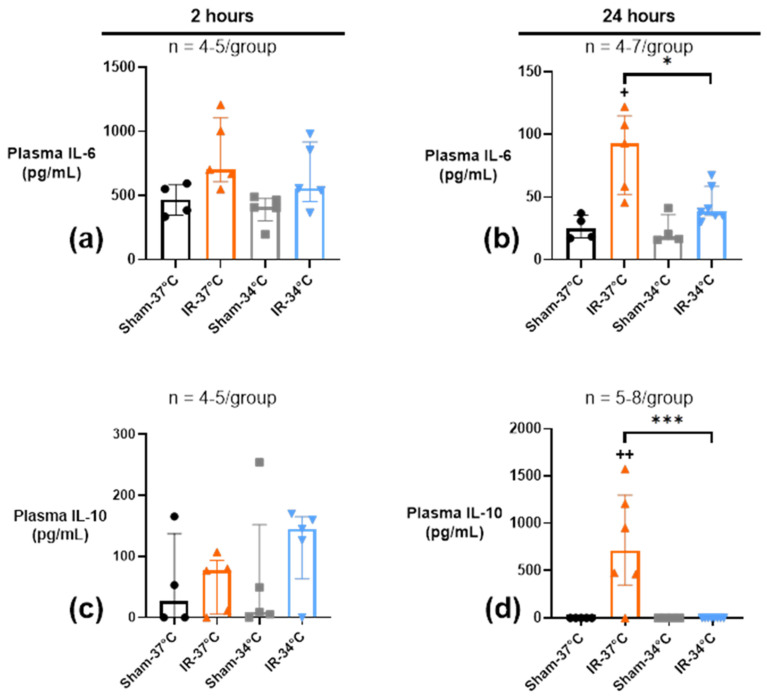
Plasmatic concentrations of inflammatory markers IL-6 (**a**,**b**) and IL-10 (**c**,**d**) after 20 min of bilateral renal ischemia where the mice body temperature was maintained at normothermia (IR-37 °C) or at mild therapeutic hypothermia (mTH) (IR-34 °C), and 2 h or 24 h of reperfusion, or after a Sham procedure with the same body temperature (Sham-37 °C and Sham-34 °C). IL-6 plasmatic level peaked 2 h after reperfusion in the IR-37 °C group. At 24 h of reperfusion, IL-6 concentration was lower but still increased in IR-37 °C, and reached a higher value than in IR-34 °C group. IL-10 plasmatic level peaked at 24 h only in the IR-37 °C group, with almost undetectable level in the IR-34 °C group. Data are shown as median with interquartiles. * *p* < 0.05, *** *p* < 0.001, Mann–Whitney test. + *p* < 0.05, ++ *p* < 0.01 vs. Sham-37 °C, Mann–Whitney test.

**Figure 6 ijms-23-09229-f006:**
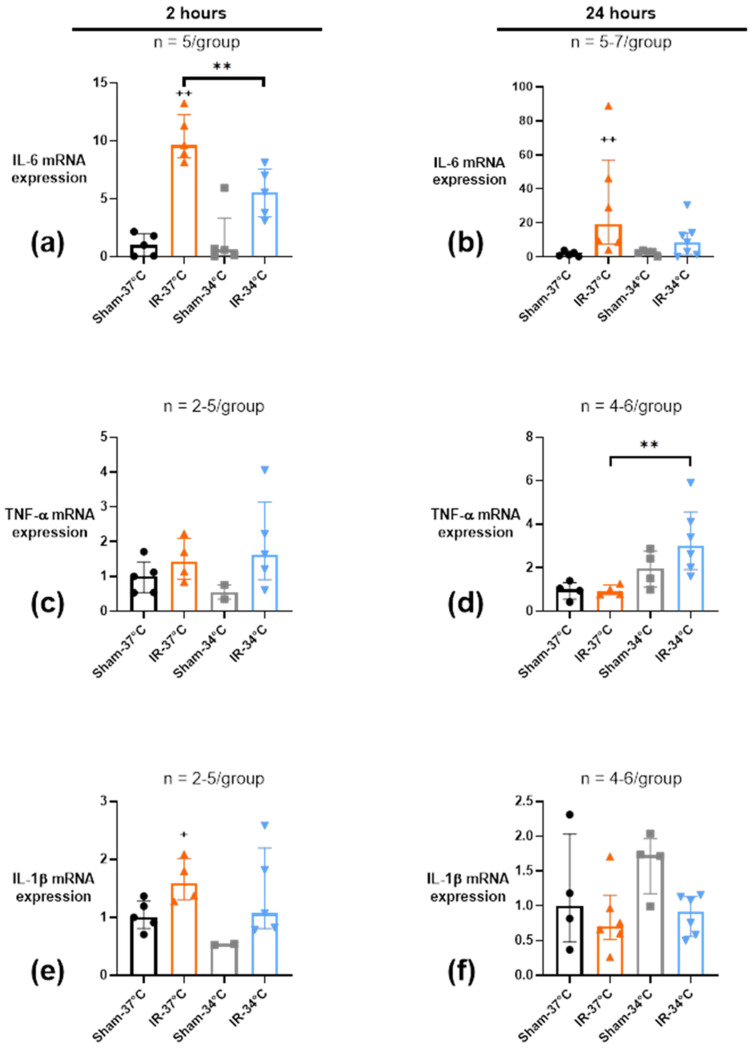
mRNA expression of inflammatory markers IL-6 (**a**,**b**), TNF-α (**c**,**d**) and IL-1β (**e**,**f**) in kidney tissue after 20 min of bilateral renal ischemia where the mice body temperature was maintained at normothermia (IR-37 °C) or at mild therapeutic hypothermia (mTH) (IR-34 °C), and 2 h or 24 h of reperfusion, or after a Sham procedure with the same body temperature (Sham-37 °C and Sham-34 °C). Renal IR induced an overexpression of IL-6 and IL-1β after 2 h of reperfusion, that was maintained after 24 h only for IL-6. However, mTH during ischemia prevented them, but induced the overexpression of TNF-α 24 h after reperfusion. Actin mRNA was used as an internal standard to normalize the abundance of cDNA in each sample to normalize results. Data are shown as median with interquartiles. ** *p* < 0.01, Mann–Whitney test. + *p* < 0.05, ++ *p* < 0.01 vs. Sham-37 °C, Mann–Whitney test.

**Figure 7 ijms-23-09229-f007:**
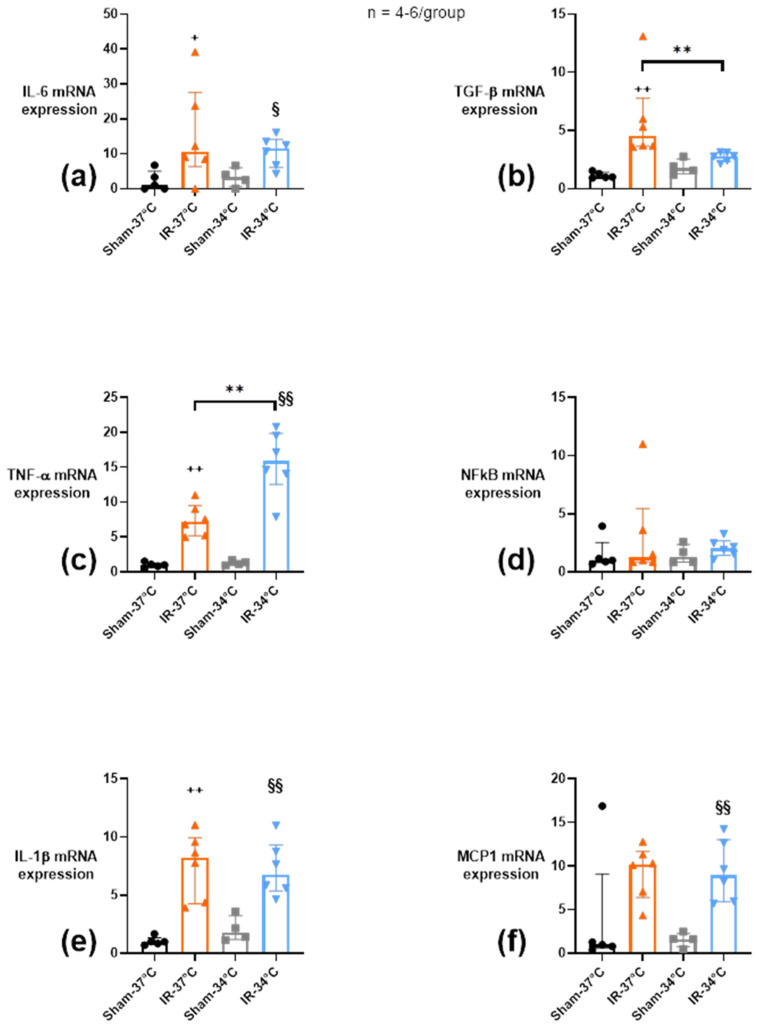
mRNA expression of inflammatory markers IL-6, TGF-β, TNF-α, NFkB, IL-1β and MCP1 assessed by qPCR in ischemic kidney after 15 min of unilateral renal ischemia where the mice body temperature was maintained at normothermia (IR-37 °C) or at mild therapeutic hypothermia (mTH) (IR-34 °C), followed by 1 month of reperfusion, or after a Sham procedure with the same body temperature (Sham-37 °C and Sham-34 °C). (**a**–**f**) Renal ischemia-reperfusion induced an overexpression of IL-6, IL-1β, TGF-β, TNF-α, and a trend toward MCP1 after 1 month of reperfusion. mTH applied during ischemia however attenuated the chronic overexpression of TGF-β, but increased it for TNFα. Actin mRNA was used as an internal standard to normalize the abundance of cDNA in each sample to normalize results. Data are shown as median with interquartiles. ** *p* < 0.01, Mann–Whitney test. + *p* < 0.05, ++ *p* < 0.01 vs. Sham-37 °C, Mann–Whitney test. § *p* < 0.05, §§ *p* < 0.01 vs. Sham-34 °C, Mann–Whitney test.

**Figure 8 ijms-23-09229-f008:**
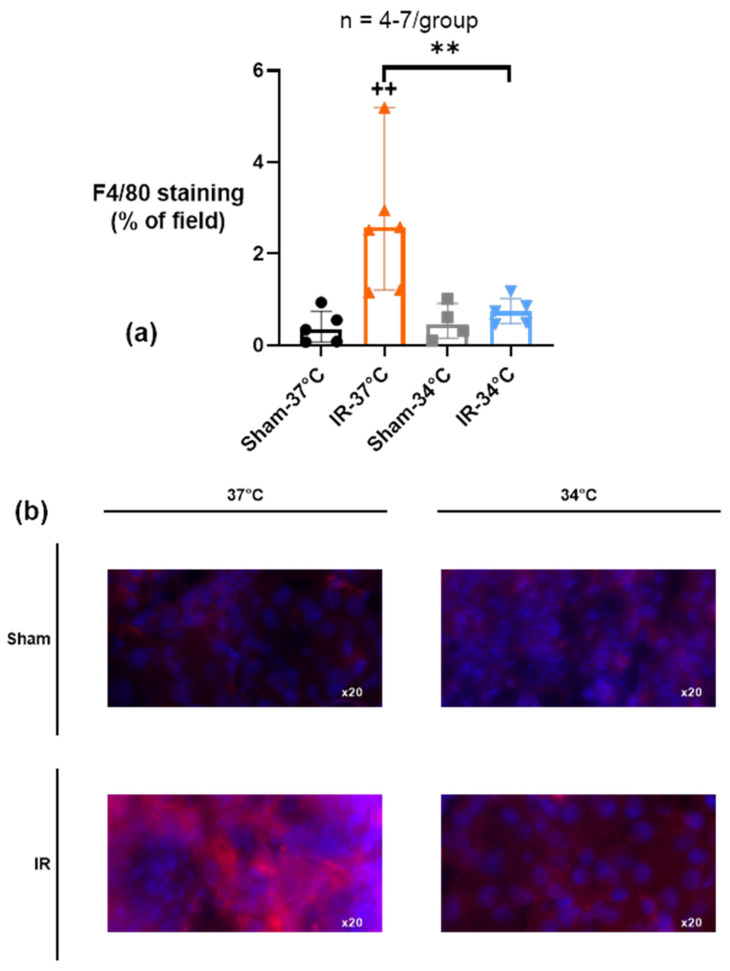
(**a**) F4/80 staining (red) assessed by immunofluorescence in ischemic kidney after 15 min of unilateral renal ischemia where the mice body temperature was maintained at normothermia (IR-37 °C) or at mild therapeutic hypothermia (mTH) (IR-34 °C), followed by 1 month of reperfusion, or after a Sham procedure with the same body temperature (Sham-37 °C and Sham-34 °C). After 1 month, renal ischemia-reperfusion induced a significant infiltration of macrophages in the kidney tissue, which was prevented by mTH. (**b**) Representative images of 8-µm sections of immunostained kidney. Nuclei were stained with DAPI (blue). Data are shown as median with interquartiles. One data point in the IR-37 °C group is outside the axis limit (value of 13.07) for better vizualisation. ** *p* < 0.01, Mann–Whitney test. ++ *p* < 0.01 vs. Sham-37 °C, Mann–Whitney test.

## Data Availability

The data presented in this study are available on request from the corresponding authors.

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
