# Peer review of "Mild Therapeutic Hypothermia Protects from Acute and Chronic Renal Ischemia-Reperfusion Injury in Mice by Mitigated Mitochondrial Dysfunction and Modulation of Local and Systemic Inflammation"

_ijms, 2022, doi:10.3390/ijms23169229_

Round 1

Reviewer 1 Report

The aim of the paper by schleef et al. is to show that mild hypothermia is able to protect renal function against an episode of ischemia-reperfusion (IR). The authors used 2 mouse models of IR and investigate the effects of hypothermia on a severe acute kidney injury and on a chronic effect on renal function through functional analysis, a study of the mitochondrial function, and of the inflammatory and fibrotic processes associated with these stresses.

The paper is well written and demonstrates the positive effect of mild hypothermia on both short- and long-term IR injury. This is a relevant protocol because it reflects more accurately clinical situations such as surgical procedures requiring a renal or aortic vascular clamping.

In the meantime, I have some remarks to do, some of them deserve explanations or modifications.

1)    In this kind of study, the analysis of the recovery of the renal function is essential and is only assessed by the urea plasma level. it would perhaps have been necessary to measure other markers of the renal function itself. Even if this marker is pertinent, I would be interested in the measure of other parameters such as creatinine and early markers such as NGAL and/or KIM-1 or even sodium or glucose reabsorption.  These markers would reflect the functional state of the proximal tubule which is the main target in case of I/R stress and oxygen depletion.

2)    I suggest mixing figure 2 with figure 1 since it concerned the same kind of data on tissue alteration.

3)    In figure 3a it appears that "sham" animals are divided into control and ischemic kidneys. This is not possible given the fact that "sham" animals did not undergo IR. It would be more accurate to speak about non-ischemic right and left kidney.

4)    In Figure 3a histogram clearly shows renal atrophy of the ischemic kidney lessened by hypothermia. On another hand, the authors say page 3 lines 134-137 that the contralateral kidneys developed compensative hypertrophy. I don't see that on the histogram. The length of both 34 and 37°C kidneys appears to be similar to their respective "sham" values. Even if compensative hypertrophy is a well-known phenomenon in unilateral nephrectomy it is not evident in the present case where the injured kidney can recover a large part of its original function.

5)    The representative images shown in figure 3d are of poor quality and do not bring valuable information. They have to be enhanced or removed since the information is given in figure 3b. At least they can be shifted to the supplemental data section.

6)    Figure 4 could be mixed with Figure 3 after images will be shifted in the supplemental data section.

7)    Page 5 lines 206-209 authors observed that cytokine IL-6 peaked in the IR-37°C group and that this is not the case for the 34°C group. This is not true: IL-6 peaked in the 34°C group as well as in the 37°C one. In the 2 cases, this is not significant as specified.

8)    Here again the conclusion is overestimated (page 5 line 223). In Figure 8a the elevation of IL-6 mRNA expression is not abrogated in IR-34°C, it is just lessened by about 50% compared to IR-37°C.

       9)  The images in figure 10 should be improved. They reflect the images that were used for scoring (they are likely the better ones I presume…how are the others!). The histogram will be more convincing with more detailed staining. Paraffin-embedded kidney tissue would be a better choice for F4/80 staining instead frozen sections.

Reviewer 2 Report

The authors made significant improvements to the paper. |
With new clarifications and new set of data, the authors were able to answer the concern.

Author Response

We thank the reviewer for his suggestions to improve our work.